# Relationship between Tumor Infiltrating Immune Cells and Tumor Metastasis and Its Prognostic Value in Cancer

**DOI:** 10.3390/cells12010064

**Published:** 2022-12-23

**Authors:** Huan-Xiang Li, Shu-Qi Wang, Zheng-Xing Lian, Shou-Long Deng, Kun Yu

**Affiliations:** 1College of Animal Science and Technology, China Agricultural University, Beijing 100193, China; 2National Health Commission (NHC) of China Key Laboratory of Human Disease Comparative Medicine, Institute of Laboratory Animal Sciences, Chinese Academy of Medical Sciences and Comparative Medicine Center, Peking Union Medical College, Beijing 100021, China

**Keywords:** TIICs, epithelial mesenchymal transformation, extracellular matrix, angiogenesis, pre-metastatic niche

## Abstract

Tumor metastasis is an important reason for the difficulty of tumor treatment. Besides the tumor cells themselves, the tumor microenvironment plays an important role in the process of tumor metastasis. Tumor infiltrating immune cells (TIICs) are one of the main components of TME and plays an important role in every link of tumor metastasis. This article mainly reviews the role of tumor-infiltrating immune cells in epithelial mesenchymal transformation, extracellular matrix remodeling, tumor angiogenesis and formation of pre-metastatic niche. The value of TIICs in the prognosis of cervical cancer, lung cancer and breast cancer was also discussed. We believe that accurate prognosis of cancer treatment outcomes is conducive to further improving treatment regimens, determining personalized treatment strategies, and ultimately achieving successful cancer treatment. This paper elucidates the relationship between tumor and TIICs in order to explore the function of immune cells in different diseases and provide new ideas for the treatment of cancer.

## 1. Introduction

With the rise of incidence rate and mortality, cancer is still a major problem threatening human public health safety. The occurrence of cancer is still an important factor in premature death in most countries. It is estimated that there were 19.3 million new cancer cases (excluding 18.1 million non-melanoma skin cancer) and 10 million cancer-related deaths (excluding 9.9 million non-melanoma skin cancer) worldwide in 2020. One in five people will suffer from cancer in their lifetime, and one in ten people will die of cancer [1]. The researchers predict that the number of cancer patients in the world will continue to increase in the next 50 years, and the incidence rate of all cancers in 2070 may be twice that in 2020. In addition, it is also expected that the maximum value-added is more likely to occur in countries with low human development index [2].

Tumor cells are the core elements in the process of cancer and play an important role in the occurrence and development of cancer. In addition, tumors can recruit many other cells to their surroundings and form the tumor microenvironment (TME). TME is composed of a variety of cellular and non-cellular components, and the components are complex, dynamic and constantly changing [3,4]. In addition to cancer cells, the cellular components in TME also include immune cells, endothelial cells, fibroblasts, adipocytes, etc. [3,5]. Non-cellular components are also complex, including extracellular matrix (ECM), proteolytic enzymes (especially matrix metalloproteinases (MMP), osteopontin, transforming growth factor-β and other ingredients [6]. Tumor infiltrating immune cells (TIICs) include both innate immune cells (macrophages, mast cells, natural killer cells (NK cells), dendritic cells (DCs)) and adaptive immune cells (T cells, B cells). These cells secrete signals and play an important role in the development of tumors [7,8]. Tumors in situ are usually more easily cured, and tumor metastasis is an important cause of death in most tumor patients. The process of tumor metastasis includes the invasion of tumor cells to surrounding tissues, entering the blood or lymph circulation, choosing a suitable location to colonize and continue to grow [9]. Increasingly studies have shown that TIICs play a very important role in the process of tumor metastasis (Figure 1).

In the process of tumor metastasis, multiple aspects of cooperation are needed, including the plasticity of tumor cells [10,11,12], the degradation of ECM [13,14], tumor angiogenesis [15,16] and the formation of pre metastasis niche (PMN) [17,18]. Each step is related to the success of tumor metastasis. Scientists are also trying to find a breakthrough point to block tumor metastasis. More and more studies have found that TIICs are closely related to these processes, even the whole process, so it is more important to deeply analyze the role and dynamic changes of TIICs in this process. In this article, we focus on the role of TIICs in the process of tumor metastasis, including EMT, degradation of ECM, tumor angiogenesis and PMN formation, and describe the value of TIICs in the prognosis of several tumors.

## 2. Promote Tumor Cell Invasion and Metastasis

### 2.1. EMT

EMT is a unique cell program and an important mechanism for embryonic development, tissue repair and disease [19,20]. Epithelial cells are closely connected horizontally and arranged in layers and combine with the basement membrane (BM). Therefore, they can only migrate laterally, and cannot leave the BM. After the EMT process, the adhesion molecules (E-cadherin) on the surface of epithelial cells decrease, while Vimentin and neuro-cadherin (N-cadherin) increase. These changes force epithelial cells to lose cell polarity and obtain mesenchymal cell phenotypic characteristics such as migration and invasion, anti-apoptosis and degradation of ECM [21,22,23]. Subsequent studies have shown that EMT is not a simple binary process of epithelial cells transforming into mesenchymal cells, but a complex and gradual process. It gradually transformed from epithelial cells to hybrid EMT cells (intermediate cells with epithelial cell markers and mesenchymal cell markers), and finally into free mesenchymal cells with mesenchymal markers as the main markers. The traditional epithelial markers gradually disappear with the development of transformation, while the expression of mesenchymal markers is reverse [24,25]. Cancer cells can undergo functional and morphological changes because of their plasticity. Cancer cell plasticity refers to the ability of some cancer cells to undergo dynamic transitions between different cell states [26]. Cancer cells exist in epithelioid form at the initial stage of cancer and adhere tightly to the BM. After the activation of EMT, the epithelial-like characteristics of cancer cells are inhibited and gradually transform into free mesenchymal cells, resulting in dissemination of cancer cells with self-renewal ability similar to stem cells. Such tumor cells are allowed to leave the primary tumor, providing a prerequisite for metastasis [24]. In the case of cancer cells, their plasticity allows them to express different markers and take on different functions [27,28]. It was found that epithelial tumor cells without EMT expressed epithelial cell adhesion molecules (EpCAM), and with the progression of EMT, the expression of EpCAM decreased, while CD106, CD51, CD61 changed from negative to positive. The final mesenchymal cells were characterized by CD51/61 expression or triple positivity (CD106+CD51+CD61+) [24,29]. There is complex and malignant two-way crosstalk between EMT tumor cells and TIICs [30,31]. On the one hand, tumor cells with EMT can have a direct impact on the immune system, including reducing the infiltration of immune cells with anti-tumor ability or making the anti-tumor immune response of the immune system less effective [31]. Studies have shown that in non-small cell lung cancer (NSCLC), the inhibitory immune checkpoint molecule PD-L1 (programmed death receptor-ligand 1) is highly upregulated in tumors at late stages of EMT progression [32]. In addition, overexpression of the immune costimulatory molecule B7-H3 upregulated mesenchymal phenotypic markers Vimentin and N-cadherin and promote EMT progression and tumor metastasis [33]. On the other hand, tumor cells can have a direct impact on the immune system, including reducing the infiltration of anti-tumor immune cells or increasing the number of cells that promote tumor migration, or making the anti-tumor immune response of the immune system more effective [31].

Macrophages recruited in the microenvironment of tumor tissue are called TAMs (Tumor-associated macrophages), which have been shown to acquire the M2-polarized phenotype and promote tumor development [34,35,36]. TAMs accompany almost the entire process of tumor metastasis [35]. Studies have shown that TAMs induce EMT and enhance the invasive ability of cancer cells [37]. The abundance of macrophages was assessed using the pan-macrophage marker CD68, and the results showed that CD68 expression correlated with EMT markers in breast cancer [37,38]. TAMs activate the NF-κB/SOX4 signaling pathway by secreting CXCL1, which promotes EMT and lung metastasis of breast cancer [39]. In addition, CXCL1 can activate NF-κB/Foxp3 signaling pathway, promotes the generation of Tregs and enhances the immunosuppressive intensity of TME [40]. MiR-106b is highly expressed in EMT-CRC (colorectal cancer) cells and has been identified as a key functional molecule that induces macrophages to polarize to M2 phenotype. This molecule can be transferring into macrophages to play a role in inhibiting the expression of PDCD4 in macrophages and activating PI3Kγ/Akt/mTOR signaling pathway. Thus, promoting M2 macrophage polarization [41]. Furthermore, activated M2 macrophages up-regulated FoxQ1 expression through the IL-6/STAT pathway. The upregulation of FoxQ1 accelerates the induction of EMT in tumor cells and promotes tumor metastasis and invasion [42]. In addition, TAMs are also considered to be the main source of multifunctional protein Gas6, and the activation of Gas6-Axl signaling pathway has a positive effect on EMT and tumor metastasis [43,44]. After blocking the expression of Gas6, it was found that the expression of EMT related transcription factors such as snail, ZEB2 and vimentin decreased significantly [44]. It can be seen that macrophages promote the occurrence of EMT in many aspects.

In one study, interferon-γ (IFN-γ) released by NK cells induced significant upregulation of PD-L1 on metastatic tumor cells. This may contribute to the immune escape of tumor cells from the anti-tumor effects of NK cells [45]. Furthermore, experiments on melanoma cells have demonstrated that NK cells directly contact with tumor cells and release IFN-γ and tumor necrosis factor (TNF-α), thereby inducing EMT of tumor cells [46]. Moreover, It was also found in gastric cancer that the proportion of activated NK cells was higher in the tumor cell microenvironment where EMT occurred. The poor prognosis of gastric cancer is closely related to the high expression of vimentin and the low expression of e-cadherin and EpCAM. Vinculin, a skeleton protein associated with cell-cell and cell-matrix junctions, inhibits EpCAM expression through CpG methylation and promotes EMT and tumor cell metastasis [47,48]. However, studies to the contrary have shown that the binding of NK cells to the receptor NKp46 promotes the secretion of IFN-γ. This further induces the increased secretion of fibronectin 1 and inhibits the occurrence of tumor EMT and tumor metastasis [49]. Moreover, the NK cell surface receptor NKp44 binds to the PDGF-DD subtype of the platelet-derived growth factor (PDGF) family to activate NK cells and induce tumor cell growth arrest [50]. Additionally, an interesting study has found that NK cells have no effect on primary tumors, but when tumor cells undergo EMT, the sensitivity of NK cell-mediated cytotoxicity increases. In other words, EMT-induced tumor cell metastasis may be the lure of NK cells. The occurrence of EMT makes the tumor killing effect of NK cells amplified as much as possible [51,52]. The invasive mast cells in bladder cancer are significantly more than those in adjacent non pathological tissues. It has been shown that mast cells can induce further tumor cell invasion by regulating EMT via the ERβ/CCL2/CCR2 signaling axis (Erβ, estrogen receptor β). In addition, the activation of this signal pathway can also promote the secretion of some MMP9 [53]. The role of MMPs in inducing EMT has been confirmed [54]. The latest research shows that the exosome KRT6B has a positive effect on the development of EMT, and the infiltration density of M2 macrophages and mast cells in the high expression group of KRT6B is significantly higher than that in the low expression group [55]. In thyroid cancer, stimulation of mast cell derived IL-8 can enhance the phosphorylation of Akt (protein kinase B) and induce the continuous increase in Slug, an EMT transcription factor with zinc finger structure. And ultimately promote EMT [56].

DCs secrete a high level of CXCL1 protein, which stimulates the high expression of MMP-7 and EMMPRIN and promotes EMT. This process is characterized by the upregulation of Snail and Vimentin (vimentin promotes EMT by changing cell shape and movement), and the down regulation of epithelial cell marker E-cadherin [57]. In addition, CCL5 produced by DCs promotes the development of EMT, which may be related to the upregulation of non-coding RNA MALAT-1 (metastasis related lung adenocarcinoma transcript 1) [58]. During the induction of EMT by the transcription factor Snail, immunosuppressive Tregs cells with high Foxp3 expression increased in the tumor environment [59].

### 2.2. Degradation of ECM

ECM is a complex non-cellular network composed of a variety of macromolecules around cells, mainly containing proteoglycans (PGs), collagen, elastin, laminin, and glycosaminoglycans [60,61]. According to its distribution position, it can be divided into BM and interstitial matrix. BM is a sheet deposit that can provide adhesion sites for epithelial cells. Hemidesmosomes anchor the epithelial cells to BM through transmembrane protein integrin and connect with the intermediate fibers in the cell, so that the epithelial cells stably adhere to the BM [62]. Studies have shown that PGs, collagen IV, laminin, nidogen1 and 2 are the main components of BM. Among them, nidogen can connect collagen IV and laminin to maintain the stability of BM [63,64]. The ECM is not simply an inert support for cell survival. In fact, the more important function of ECM is to transmit signals and regulate the bidirectional flow of information between extracellular and intracellular [65]. The abundant cell surface receptors (such as integrins, syndecans, and discoid receptors) integrate and transmit the information contained in ECM to the cell, thus affecting the biological behaviors of cell [65,66]. Of course, cells existing in ECM, such as epithelial cells, macrophages, mast cells and fibroblasts, also secrete macromolecular matrix into ECM under the control of cell signals to become part of the ECM and promote the structural remodeling of ECM [63].

ECM is a dynamic and complex structure that is constantly reshaped in order to maintain the stable state of the organization [67]. However, when its composition changes dramatically, it may cause the occurrence of disease or metastasis of tumor. Hynes proposed that ECM has a potential role in the occurrence and development of tumors that cannot be ignored [68]. The stiffness and degradation of ECM play a critical role in tumor progression. The composition, density, tissue structure and post-translational modification of ECM determine the stiffness of ECM [69]. In tumor tissues, cells such as immune cells and cancer-associated fibroblasts produce a large amount of MMPs and lysine oxidase to promote the remodeling of ECM, so that tumor tissues are harder than ordinary tissues. The increased stiffness of ECM leads to a decrease in the adhesion strength of cancer cells to the BM, which stimulates widespread metastasis of cancer cells. In addition, the activation of TGF-β signals leads to the occurrence of EMT [70]. The rigidity of ECM effectively blocks drug penetration to the tumor site and modifies the TME to promote tumor angiogenesis, EMT, and tumor metastasis [71]. Moreover, a study on glioblastoma cells found that the increased rigidity of ECM led to widespread spread of tumor cells and caused tumor metastasis, but when the stiffness was reduced, the efficiency of tumor cell metastasis was effectively reduced [72]. Furthermore, excessive degradation of ECM will lead to tissue destruction and promote the development of tumorigenesis. Its influencing factors include MMPs, cathepsin, bone morphogenetic protein 1, toll-like protease, hyaluronidase, and heparinase [73]. Among these, MMPs is the key hydrolytic protease for ECM degradation [71]. For example, MMP2 and MMP9 can degrade the ECM in TME and destroy the barrier against cancer invasion [74,75]. Degraded small pieces of ECM (including collagen, elastin, proteoglycans) can be used as tumor biomarkers to measure tumor activity and invasiveness [76]. Although it seems that the rigidity and degradation of ECM are essentially two opposite processes, in fact, their results are consistent, that is, they will cause further development of tumors [71].

Macrophages are one of the main components of innate immune cells, with two polarization phenotypes, M1 macrophages (mediating anti-tumor) and M2 macrophages (mediating tumor promoting effect) [77]. Monocytes are recruited around tumor tissue and differentiated into macrophages through the combination of CCR2 on its surface and chemokine CCL2 (synthesized by tumor cells or other cells), which together form TME with other components [35]. TAMs can degrade the ECM in the microenvironment of tumor cells by secreting proteolytic enzymes, so as to provide suitable environmental conditions for tumor cell metastasis [78]. Studies have found that E-cadherin can reduce the number of CTCS (tumor cells circulating in the blood) and inhibit the extravasation and metastasis of tumor cells [79]. TAMs induce EMT process of cancer cells by producing MMP-9 and activating PI3K/AKT/SNAIL signaling pathway, ultimately leading to severe metastasis of gastric cancer [75]. MMPs produced by cancer-associated fibroblasts in TME mediate the cleavage of collagen, among which MMP-14 is the most expressed. Next, small collagen fragments were internalized and completely degraded by cancer-associated fibroblasts and TAMs [80].

One study showed that the ability of T cells to degrade sulfated proteoglycans in the ECM was shown to be highly similar to that reported in metastatic tumor cells [81]. Furthermore, in vitro studies showed that activated CD4^+^ T cells stimulated fibroblast mediated collagen degradation and MMP-9 activation [82]. In addition, the levels of CD8^+^ T cells, CD4^+^ T cells, memory B cells and activated DCs in patients with high MMP-11 expression were significantly lower than those in patients with low MMP-11 expression [83]. Multiple MMP members, including MMP-1, -2, -9, -13, MT1-, MT2-, MT3- and MT6-MMP, were expressed in newly isolated human NK cells. Moreover, one study showed that NK-92 cells overexpressed invadopodia/podosomes and had high migration ability. In addition, NK-92 expresses a variety of MMPs, especially MMP-9, and leads to widespread disintegration of its culture environment (Matrigel) [84]. Once the ECM around the tumor cells is degraded, the tumor cells seem to open the closed door and are free to find new locations within the body. The role of mast cells in allergic reactions, inflammation, involvement in host defense mechanisms against parasitic infestations, and T cell-mediated immune regulation has been well studied. However, the role of mast cells in the development of tumors is still relatively limited. Studies have shown that direct contact between mast cells and T cells can activate mast cells, which then release cytokines and MMP-9 [85]. In addition, mast cell-derived trypsin activates enzymes such as MMP to promote neovascularization. But there is no denying that these enzymes also have an effect on ECM remodeling [86]. In conclusion, TIICs plays an important role in the regulation of ECM and is closely related to the occurrence and development of tumors.

### 2.3. Angiogenesis

As early as 1971, Judah Folkman put forth the view that “tumor growth depends on angiogenesis” [87]. The maintenance of tumor sustainable growth depends on blood vessels to provide enough oxygen and nutrients. In addition, angiogenesis is also conducive to tumor invasion and metastasis [88,89]. VEGF is a highly specific mitogen of vascular endothelial cells, which can induce endothelial cell proliferation and promote metastasis, and plays a central role in the regulation of angiogenesis [90]. Its expression and activity are regulated by many factors, including hypoxia inducible factor-1α, transcription factors, inflammatory mediators, etc. [91]. Drugs targeting VEGF can effectively inhibit tumor angiogenesis, thereby inhibiting tumor growth and metastasis [92]. Bevacizumab is a humanized monoclonal antibody against VEGF-A. Numerous studies have shown that it has antitumor activity. On February 26th, 2004, bevacizumab was approved by the U.S. Food and Drug Administration as a first-line therapeutic drug for metastatic colorectal cancer. It also became the first therapeutic drug approved by FDA for targeting tumor angiogenesis [93,94]. In conclusion, the use of anti-VEGF and anti-VEGFRs therapy for metastatic tumor angiogenesis is of great significance at present [95].

It has been found that in addition to being mainly derived from hypoxic tumor cells, another important source of VEGF-A in human glioblastomas is TAMs [96]. Moreover, another study found that in T-47D breast cancer and SW620 colon cancer, the secretion of VEGF by macrophages was even 2–3 times that of tumor cells. Further, the secretion of VEGF by macrophages increased multiple times under the interaction between tumor cells and macrophages [97]. In addition to VEGF, TAMs also secrete a variety of cytokines in tumors, such as PDGF, IL-8, TNF-α and basic fibroblast growth factor, which further promote tumor invasion and metastasis [98,99,100,101,102]. A new study also found that VEGF derived from decidua can promote the transformation of macrophages from M1 type to M2 type, and also has an effect on the recruitment of macrophages [103]. Another interesting study found that under anoxic conditions, TAMs enhance the resistance to anti VEGF drugs by producing angiogenesis promoting cytokines [104]. One study found that colony stimulating factor-1 (CSF-1) regulates the production of macrophages [105], enhances the response of macrophages to M2 stimulation, and promotes macrophages to polarize into M2 phenotype [106]. Thus, the use of CSF1R inhibitors appears to be able to overcome the resistance caused by anti-VEGF drug and significantly reduce the number of macrophages [107].

Through research and evaluation, IFN-γ is necessary to inhibit tumor angiogenesis and delay the growth of tumors, especially in the early growth of tumors [108,109]. In the process of IFN-γ exerting this physiological effect, IL-12 plays an important induction role during this period [110]. Subsequent studies showed that NK cells were essential mediators for IL-12 to inhibit angiogenesis. Activated NK cells can directly kill endothelial cells, an important element in angiogenesis [111]. It was later found that decidual NK cells are effective secretions of a series of angiogenic factors during pregnancy and induce the growth of blood vessels in decidua. So, do NK cells play a similar role in tumor angiogenesis? [112]. A large number of researchers are very interested in this scientific problem. It has been found in many cancers such as renal cell carcinoma, breast cancer and NSCLC. Tumor invasive NK cells can be reprogrammed, and the reprogrammed NK cells have the effect of promoting angiogenesis, thus leading to tumor invasion [113,114,115]. Therefore, the role of NK cells in promoting tumor invasion has gradually attracted people’s attention, and there are still many aspects to be explored.

Mast cells have a dual role similar to macrophages in targeting cancer. On the one hand, mast cells can release IL-4 and TNF-α to promote tumor cell apoptosis. On the other hand, chemical attractants released by tumor cells recruit a large number of mast cells to the surrounding tumor tissue and induce mast cells to release various substances that promote tumor progression, including VEGF, IL-8 and heparin, which are conducive to angiogenesis [116]. IL-8 is an angiogenic factor, which directly enhances the proliferation, survival and MMP expression of vascular endothelial cells, and promotes angiogenesis and tumor metastasis [117].

Regulatory T cells (Tregs) are CD4^+^ CD25^+^ T cell subsets and are believed to promote tumor development mainly through immunosuppression [118,119]. The study found that pathological angiogenesis was repaired with the increase in Tregs recruitment in the retina [120]. In patients with endometriosis, TGF-β1 secreted by Tregs promotes the expression of IL-8 and VEGF in primary human endometrial stromal cells through activating the p38/ERK1/2 signaling pathway, thereby promoting angiogenesis [121]. With the deepening of research, the role of Tregs in tumor angiogenesis has also attracted much attention. A study showed that the micro-vessel density of tumor increased with the increase in the number of Tregs infiltrated by tumor. In lymphocytic leukemia, the continuous accumulation of Tregs makes cancer cells penetrate into bone marrow faster and promotes angiogenesis through VEGF-A/VEGFR2 pathway [122]. The surface of Tregs can express CCR4, which is a ligand for a variety of chemokines, including CCL17 (mainly expressed in thymus) and CCL22 (mainly secreted by macrophages and DCs) [123]. CCL17/CCL22 combines with CCR4 expressed by Tregs to promote the recruitment of Tregs near the tumor [123]. Besides that, the recruitment of Tregs to tumors under hypoxic conditions is also through the combination of CCL28 and CCR10 [124].

Research on the effect of B cells on tumors has been deepening. In terms of anti-tumor function, B cells can directly present tumor-related antigens to T cells to kill tumor cells, or actively secrete antibodies that can recognize tumor antigens [125]. Therefore, the development of tumor can be effectively controlled, which is beneficial for clinical treatment. But the role of B cells in cancer is also being explored. It has been found that immune complexes induced by B cells can also cause angiogenesis by activating macrophages and complement activation [126].

### 2.4. Formation of PMN

PMN is the accumulation of immune cells and ECM in the target organs before tumor cell metastasis, which paves the environment for tumor colonization, and is the key for tumor cells to successfully migrate to other tissues and organs and colonize. PMN has been proved to play an important role in the metastasis of breast cancer, pancreatic cancer and other cancers [127,128]. Various components from primary tumors, suppressor cells from bone marrow and host matrix are three important factors in the formation of PMN [18,129,130]. However, more and more studies have found that TIICs also play a crucial role in the formation of PMN [131].

During the formation of PMN, different types of immune cells will be recruited to the host organ, and abnormal differentiation and aggregation will occur [132,133,134]. Studies have shown that neutrophils may be one of the important members in the formation of PMN [134]. Neutrophils are recruited to PMN by granulocyte colony-stimulating factor (G-CSF), exosomes, and stromal cell derived factor-1 [130]. The study found that in the mouse lung metastasis model, the number of granulocytes were 5–6 times higher than in the lungs without tumor metastasis. In addition, monocytes and macrophages also had a certain amount of accumulation. Furthermore, the study also found that granulocytes were highly expressing Bv8 protein, which has previously been characterized as a pro-angiogenic factor and can directly stimulate the metastasis of tumor cells [134]. Further studies have shown that metastatic tumors overexpress G-CSF, which mobilizes granulocytes and promotes their arrival in other organs or tissues to form a microenvironment suitable for tumor cells [134]. Inhibiting the expression of G-CSF or Bv8 has a significant inhibitory effect on tumor metastasis [134].

Macrophages also play a role in the formation of PMN [135,136,137]. Tumor derived exosomes polarize macrophages in PMN into M2 phenotype with immunosuppressive function through metabolic reprogramming. During this process, the expression of PD-L1 on the surface of macrophages increases, which is mediated by two mechanisms: one is the enhancement of NF-κB dependent glycolysis, the other is the increase in the transformation of pyruvate to lactic acid, and lactic acid is fed back to NF-κB further increased the expression of PD-L1 [138]. Cytochrome P4504A (CYP4A) is significantly increased in invasive breast cancer and malignant melanoma. Researchers have proved that CYP4A protein is mainly expressed by M2 macrophage in TME [139]. One study has found that compared with wild-type macrophages and negative control macrophages, macrophages overexpressing Cyp4a10 significantly increased the number of VEGFR1 positive myeloid (Clinical markers of PMN [140]) and the levels of MMP-9, fibronectin, S100 calcium binding protein A8, thus actively promoting the formation of PMN [139]. Studies have shown that the physical remodeling of matrix by macrophages can promote tumor metastasis. During the recruitment of macrophages to PMN, micro-tracks are formed, so that tumor cells can quickly transfer to target organs along this track [141]. In addition, the release of lipid droplets related to the infiltration of a variety of immune cells, including macrophages and DCs also plays an important role in the formation of the pre-metastasis position, such as inducing immunosuppression and angiogenesis of PMN [142].

In addition to the above, other immune cells also play an important role in the formation of PMN. Studies have shown that the exosomes secreted by tumor cells can be absorbed by mast cells, thereby inducing the activation of mast cells. Trypsin released by mast cells promotes the proliferation and migration of human umbilical vein endothelial cells by activating JAK-STAT signal pathway, which helps to provide friendly conditions for tumor metastasis [143]. Ana Carolina Monteiro et al., revealed the role of T cells in inducing bone metastasis in mouse breast cancer. Before bone metastasis of breast cancer, tumor cells induce CD4^+^ T cells to express pro-osteoclastogenic cytokines IL-17F and RANKL (Receptor Activator of Nuclear Factor-κB Ligand) and promote the occurrence of bone loss [144]. In this T cell environment, DCs transform into osteoclasts and promote bone consumption [145]. These results suggest that CD4^+^ T cells contribute to the formation of PMN before tumor metastasis.

In conclusion, TIICs is closely related to the occurrence and development of tumors. Specifically, it plays an important role in a series of processes such as the transformation of tumor cells from epithelial phenotype to mesenchymal phenotype, remodeling and degradation of ECM, formation of tumor neovascularization, or the creation of suitable living conditions before the colonization of metastatic tumor cells into new tissues. Therefore, in the diagnosis, treatment and prognosis of cancer, TIICs should be paid sufficient attention. With the deepening and refinement of research, immune infiltration may become a good prognostic indicator, and immune cell markers are becoming one of the factors used to determine the prognostic value of cancer. Next, in this manuscript, we also focused on exploring the prognostic value of TIICs in several cancers and revealing its important significance in tumor therapy (Table 1 shows the relationship between TIICs and several cancers).

## 3. Prognostic Value in Tumors

### 3.1. Cervical Cancer

Cervical cancer is the most common gynecological malignant tumor disease. In 2020, the number of new cases will reach 600,000, and the number of new deaths will exceed 340,000. It is the main cause of cancer deaths in 36 countries represented by Malawi [168]. The incidence of cervical cancer in developing countries is significantly higher than that in developed countries. This is often because the HPV screening, treatment, early diagnosis guidelines and palliative treatment for cervical cancer in more backward countries are far behind other more developed countries [169]. Human papillomavirus (HPV) is the main factor leading to cervical cancer. Therefore, HPV vaccination has become an important and effective means to prevent cervical cancer [170]. At present, the treatment of cervical cancer is mainly surgical treatment and radiotherapy, supplemented by drug treatment [171]. Of course, preventing cervical cancer from the source is still the most ideal goal.

Macrophages play an important role in the prognosis of cervical cancer and may even be used as a prognostic indicator independently of other TIICs. The number and density of its infiltration are closely related to the stage of tumor and the survival time of patients [150]. It is known that the molecular marker of pan-macrophage is CD68, and M2-like macrophage is CD163 and CD23 [172]. A study showed that the density of CD68^+^ macrophages (*p* = 0.0095) and CD163^+^ M2 macrophages (*p* < 0.0001) was significantly correlated with PD-L1, which inhibits T cell response to tumor cells by binding to its ligand. Further, the high density of CD163^+^ M2 macrophages was associated with the short survival of cervical cancer patients [173]. Therefore, M2 macrophages as a key factor in promoting cancer progression, its marker CD163 has a higher prognostic value than the pan-macrophage marker CD68.

In the latest study, In the latest study, using data from 222 patients with cervical squamous cell carcinoma from TCGA, researchers found that mast cells, NK cells, and CD4^+^ T cells were statistically different from normal tissues. It was found that mast cells, NK cells and CD4^+^ T cells were statistically different from normal tissues. Whether analyzing the data from TCGA or clinical patient data, the high density of mast cells is highly correlated with the poor prognosis of patients. The survival rate of patients with high density infiltrated mast cells is about 10–30% lower than that of patients with low density infiltrated mast cells. These results indicate that mast cells are involved in the progression of cervical cancer and are potential biomarkers for the prognosis of cervical cancer, and the density of mast cells has a high prognostic value [151].

DCs have more anti-tumor characteristics in the development of cervical cancer. Clinical results show that the invasion ratio of DCs is negatively correlated with the stage of cervical cancer, and the proportion of DCs is higher in the invasive microenvironment of cervical cancer with lymph node metastasis [149]. The latest research shows that the number of CD1a positive DCs in malignant tissues is less than that in benign tissues [174]. Therefore, the antitumor activity of DCs in cervical cancer has attracted people’s attention. For example, Nocardia cell wall skeleton enhances the antitumor activity of DCs, increasing the secretion of IL-6 and IL-12, and inhibiting the expression of PD-L1, thus inhibiting the further development of cervical cancer [175]. DCs based immunotherapy and DCs related vaccines have also attracted much attention [176,177].

Wang et al., obtained data from TCGA data portal, counted 22 immune cell subtypes of 205 squamous cell carcinomas and examined the survival rate. It was found that eosinophils, activated mast cells and activated NK cells were related to poor survival rate [153]. Zou et al., made a comprehensive evaluation of the prognostic value of TIICs in cervical cancer. 406 patients with cervical cancer participated in this evaluation as the analysis object. The results showed that the high density of CD3^+^ T cells, CD4^+^ T cells, CD8^+^ T cells, CD68^+^ M cells and CD163^+^ M2 cells was significantly related to the poor prognosis (*p* < 0.001), and CD20^+^ B cells and CD57^+^ NK decreased with the progression of cervical cancer. The immune prediction model based on this analysis is also considered to be effective [152]. In addition, in the analysis of TIICs in 1290 patients with squamous cell carcinoma, immature B cells, CD4^+^ T cells, CD8^+^ T cells, mast cells, follicular helper T cells, M1 macrophages, M0 macrophages, neutrophils, eosinophils have certain clinical value for the prognosis of squamous cell carcinoma [178].

A large number of data analysis shows that the prognosis of cervical cancer is related to a variety of TIICs, such as M2 macrophages, T cells and mast cells. However, as far as the current research data are concerned, the analysis method is still general, and many research reports do not include age, duration of disease, type of cervical cancer, tumor metastasis and other indicators into the classification scope. Therefore, the accurate and specific prognostic significance of TIICs in cervical cancer needs further systematic, comprehensive and in-depth research.

### 3.2. Breast Cancer

Breast cancer is the main cause of cancer death in women, and metastatic and recurrent breast cancer has become a powerful killer, seriously threatening the safety of women’s lives [1]. Current treatment methods include surgery, chemotherapy, radiotherapy, endocrine therapy and targeted therapy. A large number of studies have shown that TIICs have important significance in the prognosis of breast cancer [179]. The distribution proportion of immune cells in different subtypes and grades of breast cancer is different, but there are more TAMs in the whole. In breast cancer patients, the increase in memory CD4 T and plasma cells seems to be conducive to the prolongation of survival (disease-free survival or overall survival), while the increase in M2 macrophages, activated NK cells, Tregs and activated mast cells is related to the decrease in survival [179,180]. Disseminated tumor cells (DTCs) and CTCs are the culprits of metastasis and recurrence [181,182]. In breast cancer patients, the increase in CTCs is correlated with increased Tregs, and is weakly negatively correlated with CD8^+^ T cells [183]. TIICs are important participants in the tumor process. Whether they promote or drag down the development of tumors, their role should not be underestimated [184]. Actively paying attention to the prognostic value of TIICs in tumors may be helpful to the diagnosis and further treatment of patients.

### 3.3. Lung Cancer

According to the latest statistical results, it is estimated that in 2020, there will be more than 2.2 million lung cancer cases worldwide, and nearly 1.8 million deaths, accounting for 18% of the global cancer deaths, ranking first. In economically developed countries, the incidence rate of lung cancer is higher [2,168]. The occurrence and development of lung cancer not only depends on the lung cancer cells themselves, but also has a close relationship with the TME and the TIICs in the environment [185]. Studies have shown that the TME of lung cancer patients contains many types of immune cells. In lung adenocarcinoma, memory B cells were associated with a good prognosis [148]. It has been shown that B lymphocytes have antigen-presenting function. In the lung cancer microenvironment, B lymphocytes can effectively present antigens to CD4^+^ T cells, thereby inhibiting the progression of cancer [186]. In addition, although M1 and M2 macrophages showed no correlation with the prognosis of lung adenocarcinoma, M0 macrophages were unfavorable to its prognosis. In lung squamous cell carcinoma, neutrophils are associated with poor prognosis [148,187]. In NSCLC, the increased infiltration abundance of CD3 and CD8^+^ T lymphocytes is associated with a good prognosis [147]. In a meta-analysis, invasive immune cell components have prognostic value in early-stage, surgically resected NSCLC. Similar to the previous results, CD20^+^ B cells, CD8^+^ T cells and NK cells are associated with a good prognosis. On the contrary, CD68^+^ macrophages and foxp3^+^ Tregs are associated with a shorter survival period [188].

As mentioned above, it is not difficult to find that the infiltration of various types of TIICs varies in different tumors or at different stages of the same tumor [189]. In fact, it may have something to do with the heterogeneity of cancer. Phenotypic and functional heterogeneity can occur among cancer cells within the same tumor due to genetic changes, environmental differences, and the plasticity of cancer cells [190]. This often leads to inconsistent or even contradictory results in the prognostic studies of TIICs in cancer. As shown in Table 1, the same kind of cells play different or opposite roles in different tumors, which causes great obstacles and difficulties for actual research. But because of this, it is still necessary to carry out more thorough research on malignant tumors. TIICs are always accompanied by the whole process of tumor metastasis, and play a role in the degradation of ECM, the generation of blood vessels around the tumor, the EMT process and the formation of the PMN. With the continuous development of medical technology, although the treatment level and conditions of various cancers have improved, they still face many treatment failures and poor treatment effects. The prognosis is of great significance for the judgment of treatment effects and further treatment. It can be used to further stratify patients, optimize personalized treatment plans, improve treatment efficiency and enhance treatment effect. The prognostic value of TIICs in general cancer has been known through some research data, but most of these data are limited to the relationship between TIICs infiltration and survival rate of patients, while the relationship between various prognostic indicators such as complications, disability, deterioration, recurrence, remission, migration and quality of life and TIICs is rarely reported. Therefore, the application of TIICs to evaluate prognosis in clinical practice is limited at present. In the future, it is necessary to determine the exact prognostic value of different TIICs for cancer by considering the heterogeneity of cancer and using more clinical data, more refined classification, and more diversified prognostic indicators.

## 4. Concluding Remarks

Tumor cells have strong adaptability, and the possibility of mutation is greater than that of normal immune cells [191]. Therefore, targeted therapy of TIICs is a very promising direction. TAMs, NK cells, T cells, mast cells and other cells are important components of TME, and they are closely related to every link of tumor metastasis (as mentioned above). For these TIICs that can promote tumor metastasis, corresponding targeted therapies can be developed, such as inhibiting their recruitment or survival, blocking factors and pathways associated with pro-tumor function, etc. [192]. In terms of targeted therapy of related cells, the research on TAMs is much more advanced than other cells. For example, the antitumor agent dequalinium-14 can inhibit the motility of macrophages, thus reducing the infiltration of macrophages in TME [193]. Another example is that IFNγ and celecoxib can inhibit the growth of lung tumors by regulating the M2/M1 macrophage ratio in TME [194]. With the deepening of research on TAMs, it is believed that more breakthroughs will be found to provide new targets for cancer treatment.

For NK cells, more research is focused on their antitumor activity [195,196]. The pro-tumor function of NK cells and related targeted therapies has been developed relatively little. In the process of promoting tumor metastasis, NK cells are passive: they usually play their anti-tumor role first, and with the extension of infiltration time and increase in infiltration number in TME, they are reprogrammed, thus transforming from anti-tumor NK cells to pro-tumor NK cells [113,114,115]. In theory, after reprogramming, the biomarkers of NK cells themselves may change, new biomarkers may be produced or the original biomarkers may disappear, which enables us to identify pro-tumor NK cells from anti-tumor NK cells, so as to precisely target these cells and achieve partial therapeutic significance. In addition, we can also try to find targets or therapies that can re-educate NK cells with pro-tumor properties into normal NK cells. These may provide new directions for cancer treatment in the future.

## Figures and Tables

**Figure 1 cells-12-00064-f001:**
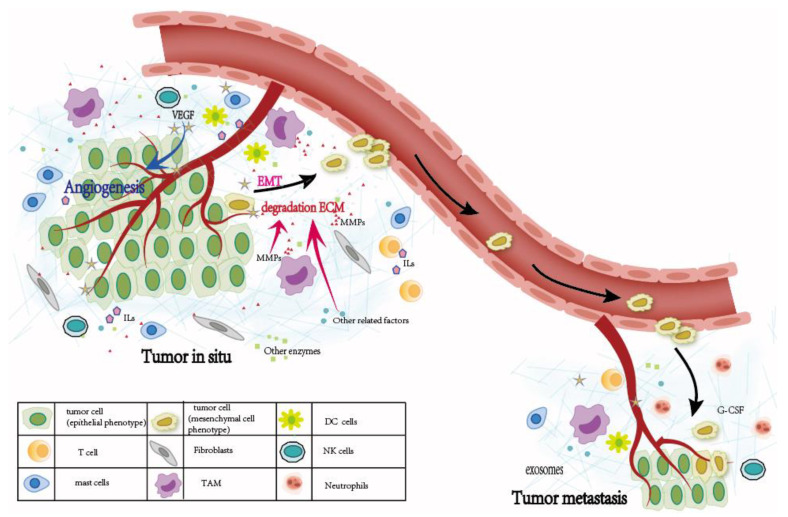
Macrophages, mast cells, NK cells, neutrophils and other TIICs are involved in all aspects of tumor metastasis, including EMT of tumor cells, ECM degradation in tumor microenvironment, angiogenesis around tumor, and the creation of microenvironment before tumor metastasis. (TAM = Tumor-associated macrophages; DCs = Dendritic cells; NK cells = natural killer cells).

**Table 1 cells-12-00064-t001:** The relationship between different TIICs and tumor prognosis.

Type of Cancer	TIICs with Good Prognosis	TIICs with Poor Prognosis	Reference
Lung cancer	B cells, CD8^+^ TIL, NK cells	M0 macrophages, Tregs	[146,147,148]
Cervical cancer	DCs, CD20^+^ B cells, CD57^+^ NK	CD68^+^ macrophages, activated mast cell, eosinophils, mast cells, NK cells, CD3^+^ T cells, CD4^+^ T cells, CD8^+^ T cells	[149,150,151,152,153]
Colorectal cancer	CD66b^+^ tumor-associated neutrophils, CD20^+^ B cells	Tregs, TAMs, monocytes	[154,155,156]
Nasopharyngeal cancer	CD8^+^ TIL, NK cells, CD4^+^ T cells, CD3^+^ T cells	Tregs, CD8^+^ T cells, neutrophils, mast cell	[157,158,159]
Gastric cancer	CD68^+^ macrophages, CD8^+^ T cells, CD3^+^ T cells, CD4^+^ T cells, M0 macrophages	Tregs, neutrophils, M2 macrophages	[160,161,162]
Biliary tract cancer	CD4^+^ T cells, CD8^+^ T cells, Foxp3^+^ T lymphocytes, mast cells	CD163 macrophages	[163,164,165]
Hepatocellular carcinoma	T cells, NK cells, DCs	Tregs, myeloid derived suppressor cells	[166,167]

## Data Availability

Not applicable.

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
