# Peer review of "Relationship between Tumor Infiltrating Immune Cells and Tumor Metastasis and Its Prognostic Value in Cancer"

_cells, 2022, doi:10.3390/cells12010064_

Round 1

Reviewer 1 Report (Previous Reviewer 2)

In this manuscript, authors presented review regarding the role of tumor infiltrating immune cells in EMT, extracellular matrix (ECM) degradation, tumor angiogenesis and the formation of premetastatic niche (PMN), and also the role of TIICs in the prognosis of cervical cancer, lung cancer and breast cancer.

The aim and scope of the study explained well. Introduction is quite comprehensive and highlighted work importance as well as its significance towards future prospective. Though, the study content now find with relevant information. However, authors have not explained the references (L53, L101-102) which authors stated as ‘error’ and commented as ‘reference source not found’. If that was the case, then authors should not tried including studies whose source was not mentioned.

Also, before proceeding further, I expect the authors to thoroughly proofread the document and fix all grammatical and typographical errors.

Author Response

Dear reviewer:

We would like to thank you for your careful reading, helpful comments, and constructive suggestions, which has significantly improved the presentation of our manuscript. We have carefully considered all comments from the reviewers and revised our manuscript accordingly. In the following section, we summarize our responses to each comment from the reviewers. We believe that our responses have well addressed all concerns from the reviewers. We hope our revised manuscript can be accepted for publication.

Reviewer 1:

  1. The misquoted references in the article have been removed and we have checked to make sure that references are properly cited in every part of the article.
  2. We have thoroughly proofread the document and corrected all grammatical and typographical errors.

Reviewer 2 Report (New Reviewer)

Li, et al.

Relationship between tumor infiltrating immune cells and tumor metastasis and its prognostic value in cancer

This review tries to summarize current understandings about the role of tumor infiltrating immune cells (TIICs) in cancer progression and metastasis. This is a highly complicated and controversial issue in cancer research. This reviewer considers that this review could be educational if it is published with revisions. Listed below are suggestions for consideration and revision.

1. Heterogeneity of the human cancer. Though the authors believe that “TIICs are important participants in the tumor process (line 479)”, whether TIICs and tumors have a causation or correlation relationship remains unclear. This confusion becomes clearer in Table 1, where T lymphocytes are said to be related to poor prognosis of cervical cancer but to good prognosis of gastric cancer, while to both good and poor prognoses of nasopharyngeal cancer. In addition, CD45+ cell infiltration in clinical tumor specimens has been said to be associated with favorable prognosis. This reviewer suggests the manuscript be revised to acknowledge tumor cell heterogeneity, so to reflect the reality of TIIC research, which is often puzzling and sometimes conflicting.

2. Preferred discussion of certain immune cells over other immune cells. At the beginning, the review specifies TIICs as all the cells of innate and adaptive immunity. Main body of the review, however, spends major efforts on the role of macrophages, dendritic cells, mast cells, and natural killer cells, without seriously elaborating the roles of B and T lymphocytes. A short discussion should be provided to clarify the involvement of B and T lymphocytes, especially when tumor infiltrating lymphocyte (TIL) amplification and CAR-T engineering are currently being proposed as effective anti-tumor therapies.

3. ECM and metastasis. The review gives seemingly conflicting observations without providing a discussion. For instance, it says that high ECM rigidity favors metastasis (lines 211 to 214), and immediately, the review starts to show that enzymatic ECM degradation has similar effect (lines and paragraphs afterwards).

4. Figures. This reviewer thinks that both Figures should be modified or otherwise removed, because the current versions do not have significant contributions to the central theme. Figure 1, for example, does not display the importance of the four aspects related to metastasis (EMT/MET, ECM degradation, angiogenesis, and PMN formation), while Figure 2 fails to show any involvement of TIICs in EMT/MET transitions.

5. Editing. Places in the manuscript that have to be revised include formatting, editing, word spacing, grammatic/typographic errors, unnecessary abbreviations, or citation issues. Some long sentences should be simplified for the reader’s consideration. Lines where these happen are listed:

Lines 16, 30, 46, 47, 53, 71, 101-102, 105, 106, 119, 127, 133-134, 147, 148-149, 151, 153, 162, 164-165, 170, 173, 174, 186, 191, 195, 218, 231, 232, 238, 239-242, 244, 265-267, 273-277, 286, 289, 291, 312, 315, 319, 320, 345, 359, 359, 380, 383, 393, 397-398, 421-423, 424, 425, 426, 427, 449, 450, 451, 455, 459, 478, 488-490, 494, 500, 501-502, 503, 504, 522-524, and 530. Similar issues are found in Table 1.

Author Response

Dear reviewer:

We would like to thank you for your careful reading, helpful comments, and constructive suggestions, which has significantly improved the presentation of our manuscript. We have carefully considered all comments from the reviewers and revised our manuscript accordingly. In the following section, we summarize our responses to each comment from the reviewers. We believe that our responses have well addressed all concerns from the reviewers. We hope our revised manuscript can be accepted for publication.

  1. According to the suggestions of the reviewers, in order to better explain the complex relationship between TIICs and tumors, we added the description of "cancer heterogeneity" in the paper. See lines 500-508.
  2. As for the problem that B cells and T cells are less described in the paper, we have added them. However, due to the lack of relevant studies on T cells and B cells in the occurrence of EMT, degradation of ECM, angiogenesis and formation of PMN, we only briefly described them in some chapters. See lines 306-328.
  3. In the section on ECM and tumor metastasis, we added an explanation of the relationship between ECM rigidity and degradation. See lines 197-202 and lines 211-213.
  4. We have deleted Figure 2 and modified Figure 1.
  5. For the convenience of readers, we have properly simplified the long sentences in the article, and checked and revised the grammar, abbreviations, citations and so on.

This manuscript is a resubmission of an earlier submission. The following is a list of the peer review reports and author responses from that submission.

Round 1

Reviewer 1 Report

The manuscript is poorly written, and needs careful editing of the entire text to improve both grammar and flow. There are sections that are very well written, followed by sentences that do not make any sense. It is difficult to review the manuscript in the current format. There appears to be formatting issues in the manuscript resulting in the merger of multiple words at multiple points in the text. Most of the paragraphs could benefit from re-structuring for better flow. There are no concluding remarks from the authors after discussing a section.  The extracellular matrix is very lightly presented, and could even be dropped out from the manuscript. Very important concepts like plasticity, immunosuppressive processes are mentioned in a few words, doing no justice to the concept. Also, many of the examples are quite lightly presented and thus some of the conclusions could be difficult to understand for the readers. In addition, the authors do not come up or suggest any common molecular pathways (other than listing a few reported ones) that could be playing a role in the tumor immune microenvironment interplay. Linked to this, the Figure 1 could also reveal specific genes or pathways that the authors think are relevant. Figure 2 seems to be out of place in the manuscript. The acronyms or abbreviations are not expanded, or expanded multiple times, or expanded incorrectly. MicroRNAs are written in multiple ways in different places. Overall, the manuscript lists important studies in this field, but lacks the intelligent hypothetical input, and as such is not recommended for publication primarily due to many grammatical and editing issues. Below are examples of sentences that do not make any sense in the review:

 “In addition, it is also expected that the maximum value-added is more likely to occur in countries with low human development index (HDI) [2].”

These cells secrete signals

In vitro studies showed that activated CD4 T cells stimulated fibroblast mediated collagen degradation and MMP-9 activation, and secreted more tumor necrosis factor  (TNF) and interleukin-6 (IL-6) than CD8+T cells [60].”

Typos like: “T The

Mast cells can release IL-4 and TNF- α Promote the apoptosis of tumor cells, but also release IL-8, VEGF, heparin and so on to promote the production of neovascularization [95].”

“the transcription factor fxop3,”- are they talking about FOXP3?

Reviewer 2 Report

In this manuscript, authors presented review regarding the role of tumor infiltrating immune cells in EMT, extracellular matrix (ECM) degradation, tumor angiogenesis and the formation of premetastatic niche (PMN), and also the role of TIICs in the prognosis of cervical cancer, lung cancer and breast cancer. However, the study content finds a lack of innovation and novelty due to already published information on the related content. Moreover, there are references (L48, L49, L76) which the authors stated as ‘error’ and commented as ‘reference source not found’. If that was the case, then authors should not tried including studies whose sources were not available.